# Polystyrene Nanoplastics in Human Gastrointestinal Models—Cellular and Molecular Mechanisms of Toxicity

**DOI:** 10.3390/ijms262311738

**Published:** 2025-12-04

**Authors:** Agata Kustra, Kamila Maliszewska-Olejniczak, Anna Sekrecka-Belniak, Bogusz Kulawiak, Piotr Bednarczyk

**Affiliations:** 1Department of Physics and Biophysics, Institute of Biology, Warsaw University of Life Sciences—SGGW, 02-776 Warsaw, Poland; agata_kustra@sggw.edu.pl (A.K.); anna_sekrecka-belniak@sggw.edu.pl (A.S.-B.); 2Laboratory of Intracellular Ion Channels, Nencki Institute of Experimental Biology, Polish Academy of Sciences, 02-093 Warsaw, Poland; b.kulawiak@nencki.edu.pl

**Keywords:** polystyrene nanoplastics, mitochondria, oxidative stress, DNA damage, ion channels

## Abstract

Plastic pollution is a growing environmental and health issue due to the increasing presence of micro- and nanoplastics in terrestrial and aquatic ecosystems. Polystyrene nanoplastics (PS-NPs) are among the most extensively studied because of their wide occurrence, physicochemical stability, and availability for laboratory research. Their nanoscale size enables interaction with biological systems at the molecular level, promoting internalization, intracellular trafficking, and potential bioaccumulation. This review summarizes current knowledge on the cellular effects and molecular mechanisms of PS-NPs, particularly in human gastrointestinal models. The gastrointestinal tract is a primary route of nanoplastic exposure, where PS-NPs can cross epithelial barriers, interact with immune and epithelial cells, and disturb cellular homeostasis. Once internalized, PS-NPs can induce oxidative stress, mitochondrial dysfunction, and dysregulation of autophagy, leading to alterations in lipid and glucose metabolism. Excessive synthesis of reactive oxygen species may trigger DNA damage, activate the ATM/ATR–p53 signaling pathway, and impair DNA repair mechanisms, thereby contributing to genomic instability. Emerging evidence also shows that PS-NPs can interact with ion channels, affecting calcium homeostasis, membrane potential, and cell viability. Overall, these findings highlight the complex and multifaceted toxicity of PS-NPs at the cellular level and underscore the need for further research to assess the long-term risks of nanoplastic exposure.

## 1. Introduction

Over the past two to three decades, the sharp rise in plastic consumption has disrupted ecosystems and hindered progress toward sustainable development goals. According to PlasticEurope, global plastic production reached 400 million tons in 2022 and increased to 413 million tons in 2023, with a continuing upward trend. This intensifies the environmental consequences resulting from human activity.

Plastics have significantly enhanced human comfort and contributed to societal development; however, their extensive use has led to severe environmental concerns [1]. Plastic waste is highly stable and resistant to natural degradation [2], persisting in the environment for 100 to over 1000 years depending on conditions and polymer type [3]. Plastic degradation in the natural environment is a slow, multi-stage process driven by physical, chemical, and biological factors. Conventional synthetic polymers, such as polyethylene (PE), polypropylene (PP), and polystyrene (PS), exhibit high resistance to environmental conditions, leading to their long-term persistence and accumulation in ecosystems [4,5]. Various physical, chemical, and mechanical methods are used to enhance plastic degradation, each contributing to the destabilization and breakdown of polymer structures. Physical degradation methods, including UV-induced photo-oxidation and thermal exposure, initiate chain scission through the formation of carbonyl groups and oxidative cleavage, resulting in the generation of mainly low-molecular-weight oligomers rather than fully depolymerized products [6]. Photocatalytic activation, such as TiO_2_-assisted oxidation, further accelerates these processes by enhancing the formation of reactive oxygen species (ROS), thereby intensifying the surface oxidation of polymer materials [7,8]. Chemical degradation mechanisms—hydrolysis, alcoholysis, aminolysis, strong oxidation, and supercritical fluids (SCFs)—play the most direct role in depolymerizing plastics into oligomers, dimers, and monomers suitable for recycling [9]. The incorporation of pro-oxidant additives, particularly transition-metal compounds, significantly accelerates oxidative cleavage of polymer chains, leading to the formation of oxidized oligomers that are more susceptible to microbial attack. Mechanical degradation, including abrasion, shear forces, and fragmentation, does not directly produce oligomers or monomers. Instead, mechanical stress increases the surface area and introduces microstructural defects that enhance oxygen diffusion, facilitating subsequent photo-oxidation and chemical reactions [10]. These mechanically induced changes promote faster environmental weathering and contribute to the more efficient progression of the overall degradation process.

An important complementary strategy for reducing plastic pollution is the development and use of biodegradable plastics, such as polylactic acid (PLA), polyhydroxyalkanoates (PHAs), starch-based plastics, and poly(butylene succinate) (PBS). Unlike conventional polymers, these materials can undergo degradation through microbial enzymatic activity, hydrolysis, or industrial composting. However, their degradation efficiency strongly depends on environmental conditions-including humidity, temperature, oxygen availability, and microbial activity-and therefore varies widely across ecosystems [11].

Plastics are widely used due to their low cost, versatility, and durability, making them one of the fastest-growing material groups [12]. Their degradation leads to microplastics (MPs, <5 mm) and nanoplastics (NPs, typically <1000 nm or <100 nm) (Figure 1) [13,14]. NPs possess a higher surface area-to-volume ratio, enabling membrane penetration, tissue accumulation, and interactions with proteins, enzymes, DNA, and gut microbiota, potentially inducing inflammation and disrupting intestinal homeostasis [15].

Plastics play a crucial role in packaging and storage within the food industry, with several types commonly used, each possessing specific properties that make them suitable for various applications. The six primary food-grade plastics include:-Polyethylene Terephthalate (PET)—often used for bottles, containers, and trays. It is the most widely recycled plastic and is valued for its strength, clarity, and resistance to impact.-High-Density Polyethylene (HDPE)—commonly used for milk jugs, juice bottles, and grocery bags. It is tough, resistant to chemicals, and is often recycled.-Polyvinyl Chloride (PVC)—used for shrink wraps, food containers, and plastic pipes. Its versatility makes it useful for a range of food packaging, but it is less common for direct contact with food due to concerns over chemical leaching.-Low-Density Polyethylene (LDPE)—found in grocery bags, bread bags, and some food wraps. It is flexible, durable, and resistant to impact.-Polypropylene (PP)—widely used for yogurt containers, straws, and microwaveable food trays. It is heat-resistant and has a higher melting point, making it suitable for hot food applications.-Polystyrene (PS)—used in disposable cutlery, foam cups, and trays. It is lightweight and inexpensive, but also one of the most controversial due to concerns over its environmental impact and potential leaching of styrene into food.

Polystyrene’s rigidity and ease of forming make it a popular choice for disposable food packaging, despite growing concerns about its environmental impact and potential health risks. Therefore, this review focuses specifically on polystyrene nanoplastics (PS-NPs), which are nanoscale plastic particles derived from polystyrene and typically measure less than 1 µm in diameter. Owing to their small size, high surface-to-volume ratio, and unique physicochemical properties, PS-NPs are capable of crossing biological barriers, interacting with cell membranes, and inducing oxidative, inflammatory, and genotoxic responses. The aim of this review is to provide a comprehensive overview of current knowledge on the mechanisms of PS-NP uptake, metabolism, and cellular effects in human models. The following sections discuss the major routes of entry into the body, metabolic processing of PS-NPs, and their impact on mitochondrial function, oxidative stress, DNA damage, and repair mechanisms. Special attention is also given to emerging evidence on the interactions of nanoplastics with ion channels, which may represent a novel and critical pathway of cellular dysregulation.

## 2. Physicochemical Properties of Polystyrene Nanoparticles

Previous studies have demonstrated that the physicochemical properties of polystyrene nanoparticles significantly influence their biological interactions. Such studies have been conducted in various models, including the gastrointestinal and respiratory epithelium as well as endothelial cells ([16,17]). Factors such as particle size and shape, surface charge, aggregation, and the formation of a protein corona can all play important roles [16,17,18,19]. From the perspective of biological activity, the size of PS nanoparticles is highly important. This applies to both their cellular uptake and the potential metabolism of PS degradation products. Studies have clearly shown that smaller PS particles (e.g., 30–50 nm vs. 100–500 nm) are internalized much more efficiently [16,20,21,22,23]. Molecular modeling studies have additionally shown that small PS, PE, and PP nanoparticles (up to 7 nm) can readily penetrate POPC lipid membranes and localize within the hydrophobic core [24]. Another key factor is the surface charge of PS-NPs, which strongly influences their cellular uptake. This is supported by studies using surface-modified PS-NPs with different functional groups that alter surface charge while keeping other variables constant. These studies demonstrated a positive correlation between zeta potential and cellular uptake ([18]). Other research has shown that in human gastric and lung cells, aminated polystyrene particles are taken up more readily and exhibit cytotoxicity at lower concentrations compared to the same-sized carboxylated or non-functionalized particles [20,23]. It has also been shown that interactions with proteins and the formation of a protein corona, such as human serum albumin or immunoglobulin gamma-1 chain C, affect the ability of nanoplastic particles to interact with the cell membrane. The protein corona reduces particle binding to neutral membranes but enhances their interactions with anionic membranes; depending on the membrane type, this results in either stiffening or loosening and disruption of lipid and protein structure [25].

Polystyrene is a very stable and highly hydrophobic polymer, which makes it remarkably resistant to biodegradation [26,27]. Plastics composed solely of carbon–carbon backbones, such as PS and PE, are particularly difficult to break down because they lack polar or hydrolysable functional groups and are therefore unreactive toward most chemical catalysts. Consequently, their degradation typically relies on thermolysis or chemically assisted pyrolysis. According to current research, the biodegradation of polymers with C–C backbones is highly complex, and complete carbon mineralization in living cells requires the coordinated action of multiple enzymes [27,28,29]. PS degradation can occur through either cleavage of the main polymer chain or side-chain scission, resulting in distinct catabolic pathways. Cleavage of the main chain produces compounds such as styrene or toluene, which can subsequently enter aromatic-compound degradation pathways. Alternatively, degradation may begin with side-chain oxidation, where monooxygenases and aromatic-ring hydroxylases are among the enzymes involved [27,28,29,30,31]. However, the degradation routes identified so far in known microorganisms remain inefficient and of low overall effectiveness. Given these structural challenges, the search for organisms capable of efficiently degrading and metabolizing polystyrene remains a key yet difficult task and is widely discussed in the literature [27,28,31].

## 3. Mechanisms of Nanoplastics Entering the Human Body

Nanoplastics can enter the human body through several primary pathways: ingestion, inhalation, and dermal absorption. Each pathway involves distinct mechanisms and potential health impacts [32].

### 3.1. Gastrointestinal Uptake of Nanoplastics

Nanoplastics (NPs) are present in the digestive tissues of vertebrates and invertebrates, including bivalves, fish, seabirds, and marine mammals, indicating widespread contamination [13]. In humans, gastrointestinal exposure is confirmed by fecal analyses that show the presence of multiple polymer types [33]. Major dietary sources include seafood, tea bags, honey, drinking water, milk, salts, and cereals, the latter due to plant uptake and vascular translocation of 50–4800 nm particles [13]. Thus, oral exposure occurs continuously. Following ingestion, digestive conditions (including acidic pH, proteases, lipases, and bile acids) promote the formation of a protein-lipid corona that modifies the NP surface properties, colloidal stability, and interactions with mucus and epithelial recognition [34,35]. The corona can also confer biomimetic features that enhance endocytic uptake [36]. The formation of the bio-corona is a highly dynamic and time-dependent process governed by competitive adsorption of biomolecules. A rapidly exchanging soft corona, composed of abundant low-affinity proteins and lipids, forms within seconds to minutes. Over time, Vroman-type competitive displacement leads to the emergence of a more stable hard corona, enriched in less abundant, high-affinity proteins, such as apolipoproteins and coagulation factors [34,35]. This transition is considered a key determinant of NP uptake and biological identity. Following cellular internalization, the corona undergoes further remodeling driven by compartment-specific conditions, such as acidic pH and proteolytic activity [37].

The intestinal mucus layer forms the first physical and immunological barrier, consisting of a dense inner sterile layer and an outer microbiota-containing layer [38]. NP penetration depends on physicochemical characteristics: smaller and positively charged particles (<100 nm) diffuse more readily, whereas larger or hydrophobic NPs are retained. Corona formation may reduce adhesion to mucins and enhance mobility. Even in dense mucus, some PS-NPs reach enterocyte surfaces in vitro [39]. Enterocytes internalize NPs through several vesicular pathways, including clathrin-mediated endocytosis (CME), caveolin-mediated endocytosis, and macropinocytosis. Recent work on PS-NPs in Caco-2 cells shows that short exposures can preferentially activate caveolin-dependent uptake and macropinocytosis, whereas prolonged exposure engages all three routes [40]. CME is the dominant uptake mechanism in polarized, differentiated Caco-2 monolayers, as demonstrated by selective inhibition with chlorpromazine and dynasore [41]. In less differentiated cells, dynamin-dependent pathways (clathrin and caveolae) and macropinocytosis contribute more strongly to uptake, and NP surface charge further modulates pathway selectivity [42]. Macropinocytosis, a non-selective uptake mechanism that depends on actin cytoskeleton remodeling, becomes particularly relevant for larger particles or when other uptake routes are saturated. Inhibition with cytochalasin D or amiloride significantly reduces NP uptake in Caco-2 models [40,42]. Advanced intestinal systems demonstrate actin-dependent transepithelial transport, indicating that macropinocytosis can facilitate basolateral translocation [43,44]. Caveolin-mediated endocytosis, involving cholesterol-rich caveolae stabilized by caveolin-1, contributes to the uptake of smaller or surface-modified nanoparticles and may support intracellular trafficking with reduced lysosomal degradation [44]. This pathway has been shown to decrease PS-NP uptake when inhibited with genistein or filipin in Caco-2 models [42,45]. Caveolin-dependent internalization complements CME and macropinocytosis, particularly for specific NP surface chemistries, and may contribute to transepithelial transport [46]. Beyond enterocyte uptake, NPs interact with intestinal immune structures. Phagocytosis by macrophages and dendritic cells contributes to deeper tissue uptake and immune surveillance [47]. M cells in Peyer’s patches efficiently translocate particles < 200 nm to gut-associated lymphoid tissue [48,49,50], where dendritic cells and macrophages capture them, linking luminal exposure to immune activation. Particles > 200 nm have limited access to classical endocytic pathways and may instead pass through transient paracellular gaps formed during physiological enterocyte shedding, a process known as persorption [50]. Persorption becomes more significant under conditions of compromised barrier integrity, including inflammation or exposure to bacterial toxins [15]. Overall, NP uptake in the gastrointestinal tract arises from multiple interacting mechanisms: corona-driven surface modification, mucus penetration, vesicular uptake (CME, caveolae, macropinocytosis), immune-associated transport (M cells, dendritic cells, macrophages), and condition-dependent paracellular passage. Bioavailability is determined by particle properties (size, charge, coating), epithelial differentiation state, and local microenvironmental conditions (Summarized in Table 1).

### 3.2. Other Exposure Routes

#### 3.2.1. Entry of Nanoplastics Through the Respiratory Tract

The respiratory tract represents an additional, although secondary, route of human exposure to nanoplastics. Indoor and outdoor air may contain airborne MPs/NPs released primarily from textiles and tire abrasion, resulting in an estimated inhalation of ~11 particles per hour [53,54]. Deposition within the airways depends on aerodynamic diameter: particles > 10 µm are efficiently removed by mucociliary clearance, those 2.5–10 µm accumulate in bronchi and bronchioles, while particles < 2.5 µm-including nanoscale plastics-can reach the alveolar region [32,55]. In the lower airways, nanoplastics interact with epithelial cells (type I/II pneumocytes, ciliated cells) and resident immune cells. In vitro models using A549 and BEAS-2B cells demonstrate that nanoplastics are internalized via clathrin-mediated and caveolae-mediated endocytosis, with reduced uptake following specific pathway inhibition [46,56]. Macropinocytosis also contributes to the uptake of small polystyrene nanoparticles, particularly ~40 nm particles [56]. Alveolar macrophages phagocytose nanoplastics and can transport them to regional lymph nodes, contributing to immune activation and systemic dissemination [47,57]. Experimental models of the alveolar barrier show that particles < 100 nm can undergo energy-dependent transcytosis across the alveolar–capillary interface (type I pneumocytes → basal lamina → endothelial cells). This has been demonstrated both in murine alveolar monolayers and advanced air–liquid interface (ALI) systems [58,59,60]. Under conditions of inflammation or oxidative stress, disruption of tight junctions increases paracellular permeability and facilitates particle leakage through the barrier [60]. Once nanoplastics enter the bloodstream, they bypass the hepatic portal system and can be distributed to secondary organs, including the liver, kidneys and brain [61]. Although inhalation is not the primary route of human nanoplastic uptake compared to the gastrointestinal tract, it contributes to overall systemic exposure and may be particularly relevant in urban and indoor environments (summarized in Table 2).

#### 3.2.2. Dermal Uptake and Cellular Response to Nanoplastics

The skin is the third route of potential nanoplastic entry into the human body (Table 3). Although the stratum corneum provides a highly effective barrier and the hydrophobic nature of plastics limits passive penetration [65], nanoplastics can enter through appendages such as hair follicles and sweat glands, as well as through areas with impaired barrier function, including UV-damaged skin. Experimental models show that 20 nm polystyrene NPs accumulate efficiently within hair follicles, whereas larger (200 nm) particles do not reach the dermis [66]. Similar findings in human skin indicate that PS-NPs of 20–200 nm penetrate only the upper stratum corneum to depths of ~2–3 µm [67]. In perifollicular regions, 40 nm NPs can be taken up by Langerhans cells, demonstrating immune cell involvement at penetration sites [68]. Together, these results confirm that intact skin restricts NP entry, but appendage-rich or compromised regions allow limited penetration [66,67]. Keratinocytes and fibroblasts internalize nanoplastics through classical endocytic pathways. Clathrin-mediated endocytosis contributes to PS-NP uptake in A431 keratinocytes, and sucrose-induced CME inhibition reduces internalization. Evidence from keratinocytes and ex vivo skin also suggests caveolin-dependent uptake, particularly in follicular regions [69]. After breaching the epidermis, nanoparticles may accumulate in the dermis, where immune cells such as Langerhans cells and macrophages phagocytose them and initiate localized inflammatory responses [70]. In HaCaT keratinocytes, polystyrene and polyethylene NPs (30–300 nm) rapidly bind to the cell membrane, acquire a protein corona and enter via macropinocytosis. Following internalization, particles localize to endosomes and lysosomes, indicating degradation-associated trafficking. Higher NP concentrations induce oxidative stress, increased reactive oxygen species, autophagy activation, cellular senescence and reduced proliferative capacity [71]. These findings suggest that even short-term dermal exposure—from cosmetics or environmental contact—may trigger cellular stress responses, particularly under repeated or high-intensity exposure scenarios.

## 4. Bioaccumulation of Micro- and Nanoplastics in Human Tissues, Organs, and Cells

### 4.1. Micro- and Nanoplastics in Human Tissues and Organs

Micro- and nanoplastics (MNPLs) have been detected in multiple human tissues and biological fluids, indicating that systemic exposure occurs through several entry routes. In the digestive system, their presence has been confirmed in feces, blood, and saliva. Fecal samples collected from volunteers across eight countries contained a broad range of polymers, including PA, PC, PE, PET, POM, PP, PS, PU, and PVC [33]. MNPLs measuring 700 nm–2 μm have also been identified in the blood of healthy adults [73], demonstrating that ingested plastics can cross the intestinal barrier and reach systemic circulation. Polymer fibers detected in human saliva further indicate exposure within the oral cavity and salivary ducts [74]. Autopsy studies additionally report the presence of polyethylene particles in human liver and kidney tissues, supporting systemic translocation and organ accumulation [75]. Inhalation represents another relevant source of internal MNPLs. Polymer particles and fibers have been identified in human lung tissue obtained from surgical resections and autopsies, including PAN, PE, PS, PET, PMMA, PP, PTFE, PUR, and SEBS, with particle sizes ranging from 1.6 to 23 μm [74]. MNPLs have also been found in sputum samples from patients with respiratory disease. These findings confirm that airborne particles can deposit in the respiratory tract and persist in lung tissues. Beyond the gastrointestinal and respiratory systems, MNPLs have been detected in several sensitive organs. Polystyrene, polyethylene, polyethylene terephthalate, and polyvinyl chloride particles (1–20 μm) have been identified in human post-mortem brain tissue using Raman spectroscopy and Nano-FTIR [76], suggesting that MNPLs may cross the blood–brain barrier. Microplastics have also been detected in human placentas, including PP, PE, and PVC (<10 μm), raising questions about prenatal exposure and potential developmental implications [77]. Collectively, these observations indicate that MNPLs can reach multiple organ systems—including those traditionally considered protected—supporting the concept of systemic biodistribution following environmental exposure. (Summarized in Table 4).

### 4.2. Localization of Nanoplastics in Cellular Organelles

Evidence from in vitro and ex vivo gastrointestinal models shows that nanoplastics accumulate predominantly within vesicular compartments rather than freely dispersing in the cytosol. Across Caco-2 enterocytes, HepG2 hepatocytes, human kidney epithelial cells and intestinal organoids, the most consistent sites of localization are endosomes, lysosomes and autophagosomes, as confirmed by confocal microscopy and TEM imaging [80,81,82]. This pattern reflects classical endocytic trafficking, with lysosomal overload frequently associated with autophagy activation, oxidative stress and inflammatory signaling. Nanoplastic exposure also induces robust mitochondrial dysfunction, although current evidence indicates that this results from secondary stress responses rather than direct mitochondrial uptake. Reported effects include loss of mitochondrial membrane potential (ΔΨm), increased ROS generation, altered expression of mitochondrial dynamics regulators (e.g., DRP1↑, OPA1↓) and disruption of metabolic pathways [81,83]. While TEM studies commonly show swollen mitochondria and fragmented cristae, no investigations have demonstrated nanoplastic presence inside the mitochondrial matrix or intermembrane space, suggesting that mitochondrial impairment is indirect. Similarly, several studies report activation of endoplasmic reticulum (ER) stress, including increased PERK, ATF4 and CHOP expression, indicating disruption of protein homeostasis and unfolded protein response activation. However, as with mitochondria, no TEM images confirm nanoplastic localization within the ER lumen or membranes, meaning these findings should be interpreted as functional rather than structural evidence of ER involvement. Reports of nuclear localization remain limited and inconclusive. In the vast majority of studies, nanoplastics remain restricted to the cytoplasm and perinuclear region [80,81]. Rare nuclear signals have been observed primarily under co-exposure conditions, such as combined exposure to PS-NPs and silver nanoparticles [84], making it unclear whether nanoplastics alone can traverse the nuclear membrane. Overall, current evidence indicates that in gastrointestinal models, nanoplastics primarily accumulate in endosomes, lysosomes and autophagosomes, where they trigger oxidative stress, mitochondrial dysfunction and ER stress. In contrast, direct localization in the nucleus, ER lumen or mitochondrial interior remains unsubstantiated, and conclusions regarding these compartments should be made with caution. A detailed overview is provided in Table 5.

## 5. The Impact of Nanoplastics on the Structure and Function of Cell Membranes

The cell membrane is a dynamic lipid–protein structure that acts as a selective barrier between the internal and external environments of the cell. It is responsible for maintaining integrity, regulating the transport of substances, signal transduction, and metabolic homeostasis. Its properties, including fluidity and elasticity, are determined by lipid composition, cholesterol content, and environmental conditions [86].

Nanoplastics, especially polystyrene ones, can adhere to cell membrane surfaces as a result of hydrophobic and electrostatic interactions. Another important factor is the formation of a so-called biomolecular corona, which arises in biological environments and modifies the surface properties of particles. In vitro studies under conditions simulating the gastrointestinal tract have shown that the presence of a protein corona increases the uptake of PS-NPs by THP-1 macrophages, proving that cells interact with particles modified by the environment rather than with the polymer in its original state [34,35].

The internalization of nanoplastics primarily occurs through endocytosis mechanisms as described above [56]. The degree of cell differentiation is of significant importance. In the Caco-2 model, it has been shown that mature monolayers take up PS-NPs by different mechanisms than immature cells, which has a direct impact on the permeability of the intestinal barrier [42]. Studies on intestinal and lung cell lines have also found that internalization is accompanied by functional disorders such as oxidative stress and changes in barrier integrity [80,87].

The effectiveness of nanoplastics penetration is influenced by several factors. The lipid organization of the membrane and its cholesterol content can either promote or limit the internalization process. The aggregation of particles, resulting in an increase in their effective size, makes it difficult to overcome the lipid barrier and forces the use of active uptake mechanisms [60]. A positive surface charge increases affinity for negatively charged phospholipids, facilitating internalization [69]. In the case of the skin, the stratum corneum plays an important role. It has been shown that particles with a diameter of 40 nm can penetrate CD1a^+^ epidermal dendritic cells, while larger ones (750 and 1500 nm) do not cross this barrier [70]. Penetration events have also been reported in aggregation foci and within hair follicles [66,67,72]. In the gastrointestinal tract, an additional barrier is provided by the mucus layer, whose composition and physicochemical properties are important for the diffusion and availability of nanoparticles [38,88,89].

The consequences of nanoplastics interacting with the membrane include the reorganization of the lipid bilayer, reduced fluidity and integrity, and resulting disturbances in transport and signaling. In models of intestinal and pulmonary epithelium, PS-NPs have been found to reduce barrier integrity and induce activation of inflammatory pathways, including NF-κB and MAPK, as well as modulate the STAT-ERK axis [65,90,91]. Disruptions in lysosomal function and the activation of autophagy and mitophagy have also been described [46,92]. These data indicate that interactions at the cell membrane level are a key starting point for a toxicological cascade leading to oxidative stress, mitochondrial dysfunction, and inflammatory response [71,84].

In summary, the cell membrane serves as the primary point of contact between nanoplastics and the cell, and the properties of both the particles and the membrane itself determine their subsequent intracellular fate and toxicological consequences. These interactions determine the mechanisms of internalization, fate within cell compartments, and the nature of the biological response, highlighting their fundamental importance in assessing the risks associated with nanoplastics.

## 6. Cellular Metabolism and Metabolic Reprogramming Induced by PS-NPs

Understanding how cells handle PS-NPs, including their internalization, trafficking, transformation, and clearance, is critical for assessing their biological effects and associated risks. Once internalized, PS-NPs may undergo physicochemical modifications, which can trigger metabolic disturbances. Although relatively stable, they can promote oxidative or enzymatic changes in the cellular environment, frequently inducing reactive oxygen species (ROS) generation, particularly mitochondrial ROS (mtROS), which leads to oxidative damage of lipids, proteins, and DNA [93,94]. Lipid peroxidation has been linked to ferroptosis, as shown in pig oocytes exposed to PS-NPs [94].

PS-NPs may also activate endoplasmic reticulum (ER) stress via the PERK-ATF4 pathway, contributing to altered lipid synthesis and lipid droplet accumulation in mouse liver cells [95]. Furthermore, exposure induces autophagy and affects lysosomal function; TFEB, a regulator of autophagy and lysosomal biogenesis, is upregulated in response to exposure. However, autophagic flux depends on nanoparticle surface charge—cationic PS-NPs may impair lysosomal degradation [96]. In neuronal models, PS-NPs disrupt autophagosome–lysosome fusion via the TSC2-mTOR-TFEB axis, leading to autophagosome accumulation and defective protein clearance [97]. Disturbances in lipid and glucose metabolism have also been reported. In macrophages, PS-NPs co-localize with lysosomes, disrupting PPAR signaling and lipid homeostasis [92]. In hepatic cells, exposure alters glycolipid metabolism via ROS-dependent activation of NF-κB and MAPK pathways and disrupts glucose homeostasis, especially following photoaging of PS-NPs [90].

It is worth paying attention to one more aspect. Metabolism is understood not only in terms of the molecular transformations of PS-NPs, but also in how PS-NPs affect the cell’s metabolic state. In the context of mitochondrial impairment, it has been demonstrated that PS-NPs disrupt mitochondrial function by affecting CI, leading to excessive mitophagy through the AMPK/ULK1 pathway, causing dopaminergic neuron death [98]. Also, it has been clearly shown that apoptotic changes and an increase in mTOR level depended on the size of the tested NPs, while the smallest particles caused the most significant alterations [99]. Alternatively, the activation of lipid synthesis genes (e.g., via ATF4-PPARγ/SREBP-1) leads to increased accumulation of lipid droplets. This implies altered lipid metabolism, possibly contributing to fatty liver disease, among other conditions. These results demonstrate the hepatotoxic effects of PS-NPs and clarify the mechanisms of abnormal lipid metabolism induced by PS-NPs [95]. Additionally, activated hedgehog and insulin ligands by decreased transcription factor DAF-16 mediate transgenerational nanoplastic toxicity in *Caenorhabditis elegans* [100].

## 7. Mitochondrial Dysfunction and Oxidative Stress Induced by Nanoplastics in Gastrointestinal Epithelial Models

Mitochondria are the central organelles responsible for maintaining cellular energy homeostasis by coupling substrate oxidation with ATP synthesis through the process of oxidative phosphorylation [85]. They also regulate apoptotic signaling, redox balance, and cell-cycle progression, making them highly sensitive to xenobiotics [101]. Accumulating evidence indicates that nanoplastics profoundly disturb mitochondrial integrity and function in gastrointestinal cell models (Table 6). After internalization, PS-NPs localize predominantly in the cytoplasm and perinuclear region, frequently adjacent to mitochondria and the endoplasmic reticulum, as shown in Caco-2 cells using confocal microscopy and TEM [102]. Although mitochondrial ultrastructural abnormalities—such as swollen organelles and cristae disruption—have been reported in several studies, there is still no direct imaging evidence confirming NP penetration into the mitochondrial matrix, and therefore such claims should be interpreted cautiously. One of the earliest and most consistent indicators of nanoplastic toxicity is the depolarization of the mitochondrial membrane potential (ΔΨm). In Caco-2 cells, PS-NP exposure results in a marked ΔΨm decrease, accompanied by ATP depletion, oxidative stress, and apoptosis [95,102]. Structural alterations, including fragmented and dysmorphic cristae, further impair oxidative phosphorylation [103]. A decline in ATP levels is a reproducible effect across multiple models; for instance, 24-h exposure to 80 nm PS-NPs reduces ATP concentrations in Caco-2 cells in parallel with ROS overproduction [104]. Prolonged exposure or post-exposure observation periods indicate that mitochondrial impairment may persist even after the acute phase of toxicity resolves. Both in IEC-6 cells and in vivo studies, PS-NPs interfered with BNIP3/NIX-dependent mitophagy, contributing to abnormal mitochondrial turnover, ultrastructural damage, and epithelial cell death [105]. Although these findings strongly support a mechanistic link between nanoplastic exposure and enterocyte bioenergetic dysfunction, quantitative ATP data in IEC-6 cells remain limited and require verification in the original full text. Oxidative stress is a central component of nanoplastic-induced mitochondrial toxicity. Numerous studies describe increased activity of antioxidant enzymes such as superoxide dismutase (SOD) and glutathione peroxidase (GSH-Px), along with a variable response of catalase (CAT), depending on dose and exposure duration [106]. Elevated ROS generation, lipid peroxidation, and reduced viability have been consistently documented in gastrointestinal cell lines, including Caco-2 [106]. In GES-1 gastric epithelial cells, exposure to 80 nm PS-NPs resulted in decreased SOD, CAT, and GSH-Px activity and increased malondialdehyde (MDA) levels, indicating oxidative stress-driven lipid peroxidation. DNA damage biomarkers (8-OHdG, γ-H2AX) and activation of the β-catenin/YAP axis were also observed, suggesting secondary genomic and signaling disturbances following oxidative injury [103]. These findings highlight the importance of including oxidative stress markers-such as SOD, CAT, GSH-Px, MDA, and DNA damage indicators-in toxicity assessments. Particle characteristics strongly modulate mitochondrial toxicity. Smaller particles (<100 nm) exhibit greater cellular uptake, leading to stronger ROS generation and more pronounced ΔΨm depolarization during extended exposure periods [21]. PS-NPs also promote lipid and protein peroxidation and substantial ROS elevation in human PBMCs [21,107], emphasizing that mitochondrial toxicity is not restricted to intestinal cells. Given the global prevalence of micro- and nanoplastic contamination, these findings suggest that chronic mitochondrial stress may contribute to the development of long-term disorders such as inflammatory bowel disease, metabolic dysregulation, infertility, and neurodegeneration [108]. Therefore, further mechanistic research focusing on mitochondrial endpoints—including ΔΨm, ROS dynamics, ATP levels, mitophagy markers, and mitochondrial ultrastructure—is essential for accurately assessing health risks associated with nanoplastic exposure.

## 8. Molecular Mechanisms Underlying Nanoplastic-Induced DNA Damage and Repair in Gastrointestinal Cell Lines

The adverse effects of nanoplastics are also associated with genotoxic effects, particularly in gastrointestinal cell models (Table 7). Smaller particles (e.g., 100 nm) are more efficiently internalized and cause stronger oxidative stress and DNA damage than larger ones [110,111]. In gastric epithelial cells, polymethyl methacrylate (PMMA) NPs induce dose-dependent ROS production, a key driver of DNA damage [112]. Similarly, polystyrene NPs cause mitochondrial ROS accumulation in esophageal cells, promoting inflammation and cell death [113]. PMMA PS-NPs also inhibit the non-homologous end-joining (NHEJ) pathway, leading to genomic instability, chromosomal aberrations, and the formation of micronuclei, thereby contributing to cellular senescence [113]. In Caco-2 cells, PS-NPs impair DNA repair mechanisms by downregulating genes associated with base excision repair (BER) and double-strand break (DSB) repair [114]. BER is critical for repairing oxidative single-strand lesions, which are commonly induced by NPs [115,116]. Furthermore, exposure to engineered nanomaterials, including nanoplastics, has been shown to inhibit nucleotide excision repair (NER), preventing the removal of bulky, DNA-distorting lesions [115]. Mismatch repair (MMR), essential for correcting replication errors, may also be affected, further compromising genomic integrity. Although specific studies on MMR in the context of nanoplastic exposure are limited, its role in general DNA repair mechanisms suggests its involvement in repairing replication errors induced by nanoplastic-related stress [112]. Homologous recombination and non-homologous end joining are critical for repairing double-strand breaks (DSBs) [117]. Nanoplastics, such as PS-NPs, have been shown to cause DSBs, necessitating the activation of these pathways. NHEJ, in particular, has been highlighted as a key repair mechanism, with DNA-PKcs playing a significant role in mediating this pathway [117]. Inhibition of DNA-PKcs has been shown to increase susceptibility to DNA damage and genomic instability in cells exposed to nanoplastics [112]. DNA glycosylases are involved in the repair of oxidative damage, which is a common consequence of nanoplastic exposure. The inhibition of these enzymes by ENMs, including nanoplastics, underscores their importance in mitigating oxidative DNA damage [115]. Moreover, PS-NPs induce ferroptosis in intestinal epithelial cells. This process is exacerbated by the disruption of ether phospholipid metabolism and the accumulation of polyunsaturated fatty acid–ether phospholipids [110]. Nanoplastics activate several inflammatory signaling pathways, including the cGAS-STING pathway, which is associated with aging-related inflammation [113]. Additionally, PS-NPs activate the NF-κB and ERK1/2 pathways, further contributing to inflammation and gut barrier dysfunction [118]. Moreover, PS-NPs exposure leads to increased levels of interleukin 17C (IL-17C) in a specially designed mouse model, with nuclear factor erythroid-derived 2-related factor 2 (Nrf2) deficiency in their intestines, a strain whose intestines are particularly susceptible to PS-NPs, contributing to cell death and DNA damage [119]. PS-NPs disrupt mitochondrial function and inhibit mitophagy, leading to increased oxidative stress and cell death in gastrointestinal cells [46,113]. This disruption is linked to the accumulation of damaged mitochondria and exacerbated cellular damage. PS-NPs induce lipid peroxidation and oxidative stress, which are associated with DNA damage in Caco-2 cells. This process is mediated through the activation of the PI3K/AKT/mTOR signaling pathway, which also affects fatty acid metabolism [116]. Moreover, it was demonstrated that PS-NPs promote colitis-associated cancer (CAC) by disrupting lipid metabolism and inducing DNA damage. Nanoplastics alter the composition of gut microbiota, which can indirectly contribute to DNA damage and intestinal health issues. For instance, PS-NPs exposure leads to changes in the abundance of specific bacterial species and affects the production of short-chain fatty acids (SCFAs) [46,120].

PMMA NPs cause G1 phase cell cycle arrest and increase senescence-associated β-galactosidase activity in gastric epithelial cells, highlighting their role in promoting cellular aging and DNA damage [112]. These findings highlight the multifaceted mechanisms by which nanoplastics induce DNA damage in gastrointestinal cell lines, emphasizing the roles of oxidative stress, impaired DNA repair, lipid peroxidation, inflammation, and mitochondrial dysfunction.

## 9. Interactions of Nanoplastics with Ion Channels

Ion channels are proteins that allow the flow of ions through plasma and organelle membranes and play crucial roles in cell physiology and electrical signaling between cells. Ion channels regulate signal transduction, control cell proliferation and migration, and maintain cell volume. They also participate in more specialized processes, such as muscle contraction or insulin release [121]. Mutations or changes in the function of channels lead to dysfunctions in the cardiovascular and nervous systems, as well as autoimmune and metabolic diseases [122]. Some studies indicated that NMs, specifically those ranging from 1 to 100 nm, can interact with ion channels, causing changes in channel kinetics [123] and electrical currents [124]. In addition, exposure to NMs alters cellular localization [125] and the expression of ion channel-related RNA and proteins. For example, acute apical exposure to negatively charged 20 nm polystyrene nanoparticles (N20) of human airway submucosal cells, Calu-3, has been shown to significantly impact ion transport, particularly through the activation of cystic fibrosis transmembrane conductance regulator (CFTR) chloride (Cl^−^) channels. The patch-clamp experiments showed that N20 can directly interact with the CFTR Cl^−^ channels. Polystyrene particles also activated in Calu-3 cells’ basolateral K^+^ channels, which play a role in maintaining the electrochemical gradient necessary for Cl^−^ and bicarbonate (HCO_3_^−^) secretion in airway epithelium cells [126]. However, the specific binding sites of nanomaterials and the mechanisms through which they affect channel activity remain less understood.

Recently, studies highlighted changes in ion transport and concentration in tissues of various organisms due to exposure to nanoplastics. In aquatic organisms, this phenomenon is particularly observed in the gills, which play a leading role in ion exchange. Senol et al. [127] found that exposure to polystyrene nanoplastics, combined with increased temperature, induced injury and inflammation of the gill epithelium in zebrafish, affecting their ion transport process. Other studies demonstrated that exposure to PS-NPs caused a decrease in the ion content of sodium (Na^+^), potassium (K^+^), chloride (Cl^−^), and calcium (Ca^2+^) in gill tissues of organisms such as the juvenile oriental river shrimp *Macrobrachium nipponense*, mussels *Mytilus galloprovincialis*, and Pacific white shrimp *Litopenaeus vannamei* [121,128,129]. This reduction in ion content was accompanied by a decrease in the activities of various ATPases, including Na^+^/K^+^-ATPase, V(H)-ATPase, and Ca^2+^/Mg^2+^-ATPase, with an increase in the PS-NPs concentration. Another study found that the PIEZO mechanosensitive ion channel in nematodes acts as a protective mechanism against nanoplastic toxicity, particularly dopaminergic neurotoxicity. Exposure of *Caenorhabditis elegans* to PS-NPs resulted in significant changes in locomotion behaviors and sensory perception. RNAi knockdown of pezo-1 resulted in the exacerbation of these symptoms. PIEZO helped regulate oxidative and antioxidative systems in response to nanoplastics [130]. This indicates that nanoplastics can interfere with ion channels involved in sensory and neural functions. PS-NPs exposure also had a destructive influence on the immune function in juvenile *Procambarus clarkii*, leading to lipid peroxidation and oxidative damage, and inducing apoptosis, which can affect ion transport and osmotic pressure regulation [39]. It was demonstrated that significantly decreased intracellular calcium levels and contractile force in neonatal rat ventricular myocytes (NRVMs) occurred after exposure to positively charged polystyrene nanoparticles under electrical synchronization [131]. The study’s results suggested that L-type calcium channels (LTCCs) are essential for contractility, and PS-NPs may disrupt this process. The authors proposed that PS-NPs can interfere with Ca^2+^ influx through calcium ion channels via two distinct mechanisms: (1) extracellular interaction: after adsorption to the plasma membrane, PS-NPs may bind to the extracellular domain of Ca^2+^ channels, (2) intracellular interaction: following internalization, PS-NPs may interact with the cytosolic portion of LTCCs. There is still a lack of information about the structural and functional changes in ion channels following treatment with PS-NPs, as well as the mechanisms underlying the interactions of PS-NPs with ion channels in gastrointestinal cells.

## 10. Conclusions

Nanoplastics are a new class of environmental pollutants with the potential to disrupt cellular metabolism through multiple mechanisms. Their small size allows them to interact with cells in ways that larger particles cannot, making their effects on cell function particularly concerning. Understanding the full extent of how nanoplastics influence metabolism is crucial for assessing their potential risks to human health and the environment. Polystyrene nanoplastics pose a real health threat due to their ability to cause significant cellular and systemic damage. PS-NPs can cause cytotoxicity and structural alterations in biomolecules, leading to cell death through apoptosis, necroptosis, and other pathways. They induce oxidative stress, mitochondrial dysfunction, DNA damage, and inflammation, which are critical factors in cellular damage. The surface charge and size of the nanoparticles play crucial roles in determining their cytotoxic potential. Understanding these mechanisms is essential for assessing the environmental and health risks posed by nanoplastics. PS-NPs can penetrate and accumulate in various organs, including the heart, liver, and brain, causing organ-specific toxicities such as cardiac aging and neurotoxicity. Moreover, they can disrupt the intestinal barrier, leading to conditions like subsequent systemic inflammation. Compared to other nanoparticles, PS-NPs exhibit unique toxicological profiles due to their ability to carry and interact with other environmental pollutants, potentially amplifying their harmful effects. Natural nanoparticles, such as those derived from biopolymers, generally show lower toxicity and better biocompatibility compared to synthetic nanoplastics like PS-NPs, emphasizing the need for further research and regulatory measures to mitigate their impact on human health.

Further research is needed to elucidate the long-term effects of these particles on cellular and metabolic processes, as well as their broader implications for ecosystem health.

## Figures and Tables

**Figure 1 ijms-26-11738-f001:**
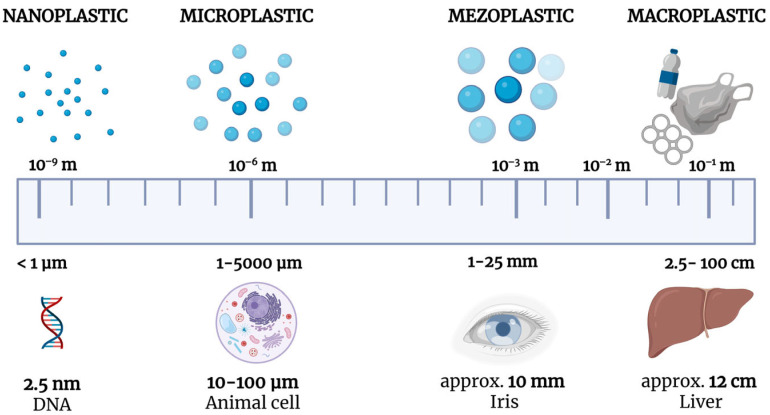
Classification of plastic particles according to their size. Plastics are commonly divided into four main categories: nanoplastics (<1 µm), microplastics (1–5000 µm), mesoplastics (1–25 mm), and macroplastics (2.5–100 cm). The scale below illustrates the relative dimensions compared to biological and anatomical structures, such as DNA (2.5 nm), an animal cell (10–100 µm), the human iris (≈10 mm), and the liver (≈12 cm).

**Table 1 ijms-26-11738-t001:** Mechanisms of internalization and translocation of nanoplastics in intestinal models.

Mechanism	Model Type	Cell Line	NP Characteristics	Exposure Conditions	Observed Effects	Reference
Clathrin-mediated endocytosis (CME)	In vitro	Caco-2	PS-NPs (≈50–200 nm; surface unmodified/COOH)	2–24 h; ±chlorpromazine	CME inhibition significantly reduces PS-NP uptake; internalized NPs localize in endosomes/lysosomes	[40]
Caveolae-mediated endocytosis	In vitro	Caco-2	PS-NPs (small < 100 nm)	±genistein or filipin	Caveolin inhibition decreases uptake of small/surface-modified NPs; caveolae support transcytosis	[42,45]
Macropinocytosis	In vitro	Caco-2	PS-NPs ~100–200 nm	±cytochalasin D, ±amiloride	Actin-dependent uptake contributes to internalization and enhances basolateral transport; inhibition reduces uptake	[40,51]
Differentiation-dependent uptake	In vitro	Caco-2 (differentiated vs. non-differentiated)	Polymer NPs (≈100 nm; +/− charge)	2–3 h	Differentiated cells rely more on macropinocytosis; undifferentiated on dynamin-dependent uptake (CME/caveolae)	[52]
Transcytosis (vesicular)	In vitro	Caco-2 Transwell/intestine-on-chip	Fluorescent NPs ~100 nm	2–24 h; static vs. flow	Shear stress increases NP translocation ~350× vs. static monolayer; energy-dependent pathway	[42,43]
Immune-associated uptake (M cells)	In vivo/ex vivo	Peyer’s patches	silica NPs (100–925 nm)	oral exposure	Particles < 200 nm efficiently transcytosed by M cells; initiate mucosal immune response	[48]

**Table 2 ijms-26-11738-t002:** Mechanisms of internalization and translocation of nanoplastics in lung models.

Mechanism	Plastic/NP Type	Size	Model Type	Cell Line/Organism	Exposure	Observed Effects	Reference
Clathrin-mediated endocytosis (CME)	PS, PLGA	40–200 nm	In vitro	A549, BEAS-2B, MDCK	4–24 h	CME inhibition (chlorpromazine) ↓ uptake; NPs accumulate in endosomes/lysosomes	[46,56,62]
Caveolae-mediated endocytosis	PS, PLGA	<100 nm	In vitro	A549, BEAS-2B	4–24 h	Caveolae inhibition (genistein/filipin) ↓ uptake; lipid-raft dependent entry	[56,62,63]
Macropinocytosis	PS	40–100 nm	In vitro	A549	4 h	Actin inhibition (cytochalasin D) ↓ uptake; macropinocytosis = major uptake route	[56,63]
Transcytosis (apical → basolateral)	PS	20–120 nm	In vitro	Mouse alveolar epithelial monolayers (MAECM)	2–24 h	Energy-dependent NP translocation across alveolar barrier; species-specific differences	[59]
Barrier-disruption dependent paracellular transport	PS	<100 nm	ALI in vitro	Human ALI lung model (cyclic stretch + inflammation)	TNF-α, ROS	TJ disruption ↑ NP passage; oxidative stress enhances leakage	[58]
Phagocytosis by alveolar macrophages	PS, PE	<200 nm	In vitro + primary cells	Alveolar macrophages	24–48 h	Uptake induces ROS, TNF-α, IL-6; high inflammatory activation	[47]
Immune transport (lymphatic dissemination)	PS	<200 nm	In vivo	SD rat model	Inhalation	NPs detected in mediastinal lymph nodes; macrophage-mediated transport	[57,64]

**Table 3 ijms-26-11738-t003:** Mechanisms of internalization and penetration of nanoplastics in the skin.

Mechanism	Plastic Type	Modification	Size	Model (In Vitro/Ex Vivo)	Exposure	Observed Effects	Reference
Macropinocytosis	PS, PE	Protein corona	30–300 nm	HaCaT keratinocytes (in vitro)	24 h	Lysosomal accumulation, ROS↑, autophagy, senescence; dominant macropinocytosis pathway	[71]
Clathrin-mediated endocytosis (CME)	PS	None	100 nm	A431 keratinocytes (in vitro)	1–6 h	CME inhibition ↓ uptake by ~40%; EGF enhances CME-dependent NP internalization	[69]
Caveolae-mediated endocytosis + immune uptake	PS	None/surface-dependent	20–50 nm	Primary keratinocytes + human skin (ex vivo)	<24 h	Rare penetration through intact barrier; uptake in keratinocytes and perifollicular tissue; partial uptake by Langerhans cells; surface chemistry influences penetration	[72]
Follicular penetration	PS	None	20 nm	Porcine skin (ex vivo)	6 h	Preferential accumulation in hair follicles; no dermal penetration	[66]
Follicular penetration (limited stratum corneum entry)	PS	None	20–200 nm	Human skin (ex vivo)	24 h	Limited entry (2–3 µm) into stratum corneum only	[67]
Phagocytosis by Langerhans cells	PS	Fluorescent	40 nm	Human skin explants (ex vivo)	24 h	Uptake by epidermal CD1a+ Langerhans cells	[70]

**Table 4 ijms-26-11738-t004:** Summary of micro- and nanoplastic occurrence in human tissues and detection techniques.

System	Human Tissue/Sample	Sample Collection	Model Type	Identified Polymers	Size	Detection Method	Reference
Digestive system	Blood	Healthy adult volunteers (n = 22)	Clinical (in vivo)	PE, PS, PP, PET, PMMA	700 nm–2 µm	μ-FTIR + Py-GC/MS	[73]
	Stool	Stool from volunteers (n = 8)	Clinical (in vivo)	PA, PC, PE, PET, POM, PP, PS, PU, PVC	50–500 µm	μ-FTIR	[33]
	Saliva	Food-contact saliva sampling cohort	Clinical (in vivo)	PE-PET-PP fibers	<100 µm	Micro-Raman + microscopy	[74]
	Liver	Autopsy liver tissue (n = 28)	Autopsy (post-mortem)	PE	1–5 µm	μ-FTIR + Raman	[75]
	Kidneys	Autopsy kidney tissue	Autopsy (post-mortem)	PE	1–5 µm	μ-FTIR + Raman	[75]
Respiratory system	Lungs	Surgically resected tissue (n = 13)	Clinical (surgical, in vivo exposure)	PAN, PE, PS, PET, PMMA, PP, PTFE, PUR, SEBS	~23 µm	μ-FTIR	[78]
	Lungs	Autopsy lungs (smokers + non-smokers)	Autopsy (post-mortem)	Polymeric particles, fibers	1.6–5.56 µm	Raman	[79]
	Sputum	Patients with respiratory disease	Clinical (in vivo)	PET, PS, PVC, PE, PAN, PU	μm range	Raman + μ-FTIR	[74]
Skin	Human skin	Ex vivo explants after NP application	Ex vivo	PS	40 nm	Confocal, CD1a immunostaining, TEM	[70]
Reproductive system	Placenta	Placental tissue from pregnancies	Clinical (in vivo)	PP, PE, PVC	<10 µm	Raman	[77]
Nervous system	Brain	Human post-mortem brain tissue	Autopsy (post-mortem)	PS, PE, PET, PVC	1–20 µm	Raman, SEM, Nano-FTIR	[76]

**Table 5 ijms-26-11738-t005:** Intracellular localization and functional effects of polystyrene nanoplastics in digestive system cell models.

Cell Model	NP Type/Size	Model Type	Methods Used	Confirmed Localization	Main Functional Effects	References
Caco-2	PS-NPs, 50–100 nm (plain, COOH, NH_2_)	in vitro	Confocal microscopy (Hoechst, LysoTracker), TEM	Endosomes, lysosomes, perinuclear region	↑ROS, oxidative stress, tight junction disruption	[52]
HT-29	PS-MPs 3–10 µm	in vitro	Confocal microscopy, TEM, flow cytometry, viability assays	Lysosomes, endosomes	Lysosomal membrane permeabilization, lysosomal stress, autophagy activation, oxidative stress, inflammatory response	[69]
HL-7702	Polystyrene nanoplastics (PS-NPs), <100 nm	in vitro	Confocal microscopy (Hoechst staining), lysosomal staining, proteomics, metabolomics, Seahorse metabolic flux analysis	Cytoplasm, lysosomes (no nuclear localization)	Metabolic reprogramming, impaired oxidative and energy pathways, ↑mTORC1 activity, mitochondrial metabolic disturbance	[81]
HepG2	PS-NPs 21.5 ± 2.7 nm	in vitro	JC-1 assay, ROS assay, TEM, LC-MS proteomics	Autophagosomes, lysosomes (no nuclear localization)	Mitochondrial dysfunction (↓ΔΨm, ↑ROS), ↑DRP1, ↓OPA1	[83]
Human intestinal organoids	PS-NPs ~50 nm	ex vivo/in vitro hybrid	Confocal microscopy, TEM, fluorescence imaging	Cytoplasm, perinuclear region, lysosomes	Differential accumulation in specific intestinal cell types; differences in uptake pathways	[82]
Caco-2 + co-exposure with Ag	PS-NPs + AgNPs, ~40 nm	in vitro	Confocal microscopy (Hoechst), TEM-EDX	Rare nuclear localization events	DNA damage, oxidative stress, genotoxicity	[85]

**Table 6 ijms-26-11738-t006:** Summary of mitochondrial membrane potential (ΔΨm), ATP, and ROS alterations induced by polystyrene nanoplastics in gastrointestinal cell models.

Cell Line/Model	Type of NPs	Size	Model Type	Dose/Exposure Time	Effect on ΔΨm	ATP Changes	ROS/Other Effects	Reference
Caco-2	PS-NPs vs. PS-MPs	80 nm; 500 nm; 3 µm	in vitro	10–100 µg/mL; 24 h	ΔΨm↓; stronger effect in presence of BPA	↓ATP (stronger for 300–500 nm MPs)	↑ROS, oxidative stress	[109]
Caco-2	PS-NPs vs. PS-MPs	80 nm; 500 nm	in vitro	100–1000 µg/mL; 24–48 h	Strong ΔΨm depolarization (NPs > MPs)	↓ATP	↑ROS, caspase activation, apoptosis	[102]
Caco-2 + okadaic acid	PS-NPs vs. PS-MPs	80 nm; 500 nm	in vitro	100 µg/mL; 24 h	ΔΨm↓; NPs potentiate OA-induced damage	↓ATP (not quantified)	↑ROS, ↑ apoptosis	[95]
PBMC	PS-NPs (NH_2_ vs. COOH)	50–100 nm	in vitro	10–100 µg/mL; 24 h	Mild ΔΨm reduction; donor variability	n/a	↑ROS, ↑ lipid peroxidation	[21]
L02	PS-NPs	~78 nm	in vitro	100 µg/mL; 24 h	ΔΨm↓ associated with metabolic disruption	↓ATP	Metabolic reprogramming, OXPHOS disruption	[104]

**Table 7 ijms-26-11738-t007:** Mechanisms of nanoplastics toxicity in gastrointestinal cell lines.

Gastrointestinal Cell Line	Model Type	Mechanism	Description	Reference
GES-1 NGEC Caco-2	In vitro	oxidative stress and ROS	increased ROS production leading to DNA damage	[112,114,116]
GES-1 NGEC Caco-2	In vitro	DNA repair inhibition	inhibition of NHEJ, HR and BER pathways, resulting in genomic instability	[114,116]
NCM460	In vitro	lipid peroxidation and ferroptosis	induction of ferroptosis through lipid peroxidation	[110]
GES-1 NGEC	In vitro	inflammatory pathways activation	activation of cGAS-STING, NF-κB, and ERK1/2 pathways	[112,117]
HET-1A HEEC IEC-6	In vitro	mitochondrial dysfunction	disruption of mitochondrial function and inhibition of mitophagy	[46,113]
Caco-2	In vitro	gut microbiota disruption	alteration of gut microbiota and increased gut permeability	[116]
Caco-2	In vitro	size-dependent effects	smaller particles induce more significant oxidative stress and DNA repair inhibition or DNA damage	[110,111,114]
GES-1 NGEC	In vitro	cell cycle arrest/senescence	induction of cell cycle arrest and senescence-associated activity	[116]

## Data Availability

No new data were created or analyzed in this study. Data sharing is not applicable to this article.

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
