# Peer review of "Polystyrene Nanoplastics in Human Gastrointestinal Models—Cellular and Molecular Mechanisms of Toxicity"

_ijms, 2025, doi:10.3390/ijms262311738_

Round 1

Reviewer 1 Report

Comments and Suggestions for Authors

The manuscript entitled “Polystyrene Nanoplastics in Human Gastrointestinal Models—Cellular and Molecular Mechanisms of Toxicity” presents a comprehensive review of the mechanisms through which polystyrene nanoplastics (PS-NPs) interact with human biological systems, particularly focusing on gastrointestinal models. The topic is of high relevance to environmental and biomedical polymer research and fits well within the scope of International Journal of Molecular Sciences. However, while the manuscript compiles an impressive volume of literature, it currently reads more like a detailed technical summary than a critical or integrative scientific review. The paper requires significant revision to improve its structure, depth of discussion, and clarity of presentation.

1.The manuscript extensively summarizes existing studies but lacks a critical synthesis of how physicochemical properties of PS-NPs influence their biological interactions. The authors should strengthen the polymer science perspective—linking polymer structure–property relationships with cellular uptake, degradation, and toxicity mechanisms.

2.While the biological mechanisms are thoroughly described, the manuscript pays little attention to the underlying polymer chemistry that governs these processes. For example, degradation kinetics, surface oxidation, and polymer chain mobility under biological conditions (pH, enzymes, oxidative species) should be discussed to explain why polystyrene behaves differently from other plastics.

3.Several sections (e.g., 2.1–2.3 and 3.1–3.2) are too descriptive and repetitive, leading to a loss of focus. The authors could merge overlapping parts into concise sub-sections and add a schematic summary of uptake and toxicity pathways. A figure summarizing “polymer property–biological effect correlations” would improve readability.

4.Tables 1–7 provide valuable information, but some lack clear differentiation between in vitro, ex vivo, and in vivo models. The authors should clearly label each study type and include nanoparticle size distributions, surface modifications, and exposure concentrations to enhance reproducibility and comparability.

5.The review should address polymer degradation to oligomers, potential bio-corona composition changes over time, and the role of copolymerized or functionalized PS derivatives. Recent developments in biodegradable alternatives and advanced detection techniques should also be included to make the review forward-looking.

Comments on the Quality of English Language

Several sentences are grammatically awkward or overly long; professional English editing is needed for fluency and conciseness.

Author Response

Response to Reviewer 1: Comments – Round 1:

We are very thankful for the in-depth review of our manuscript. The remarks made by the Reviewer are very valuable, and we believe that implementing them has greatly improved the paper. Thank you for your time and constructive feedback.

All changes in the manuscript were marked in yellow.

The manuscript entitled “Polystyrene Nanoplastics in Human Gastrointestinal Models—Cellular and Molecular Mechanisms of Toxicity” presents a comprehensive review of the mechanisms through which polystyrene nanoplastics (PS-NPs) interact with human biological systems, particularly focusing on gastrointestinal models. The topic is of high relevance to environmental and biomedical polymer research and fits well within the scope of International Journal of Molecular Sciences. However, while the manuscript compiles an impressive volume of literature, it currently reads more like a detailed technical summary than a critical or integrative scientific review. The paper requires significant revision to improve its structure, depth of discussion, and clarity of presentation.

  1. The manuscript extensively summarizes existing studies but lacks a critical synthesis of how physicochemical properties of PS-NPs influence their biological interactions. The authors should strengthen the polymer science perspective—linking polymer structure–property relationships with cellular uptake, degradation, and toxicity mechanisms.

Thank you very much for this question. Indeed, we should have paid more attention to the issues related to the structure and physicochemical properties of polystyrene and how they influence its uptake, degradation, and toxicity. We have added a brief comment addressing this point. This issue was added as a new section: “2. Physicochemical properties of polystyrene nanoparticles”.

  1. While the biological mechanisms are thoroughly described, the manuscript pays little attention to the underlying polymer chemistry that governs these processes. For example, degradation kinetics, surface oxidation, and polymer chain mobility under biological conditions (pH, enzymes, oxidative species) should be discussed to explain why polystyrene behaves differently from other plastics.

Thank you for pointing out the need to address this issue in more detail. We have added several pieces of information related to this topic. This issue was added as a new section: “2. Physicochemical properties of polystyrene nanoparticles”.

  1. Several sections (e.g., 2.1–2.3 and 3.1–3.2) are too descriptive and repetitive, leading to a loss of focus. The authors could merge overlapping parts into concise sub-sections and add a schematic summary of uptake and toxicity pathways. A figure summarizing “polymer property–biological effect correlations” would improve readability.

We fully agree with the Reviewer’s suggestion and have substantially revised Sections 2.1–2.3 and 3.1–3.2 to improve focus and eliminate redundancy. Overlapping descriptions were merged into concise, thematic subsections, and repetitive, mechanistic explanations were removed. In addition, Section 2.1 was expanded to include a more detailed discussion of biomolecular (protein–lipid) corona formation, highlighting its role in modifying nanoplastic surface properties, mucus interactions, and cellular uptake.

Instead of introducing an additional schematic figure, we chose to systematically reorganize and expand Tables 1–7, which now comprehensively summarize uptake mechanisms, toxicological pathways, polymer properties, and biological effects in a structured and comparative format. We believe that, following this revision, the tables now fulfill the intended function of a schematic summary of uptake and toxicity pathways and polymer–effect correlations.

  1. Tables 1–7 provide valuable information, but some lack clear differentiation between in vitro, ex vivo, and in vivo models. The authors should clearly label each study type and include nanoparticle size distributions, surface modifications, and exposure concentrations to enhance reproducibility and comparability.

We thank the Reviewer for this highly constructive comment. All Tables 1–7 have been systematically revised to explicitly indicate the experimental model type (in vitro, ex vivo, in vivo, clinical/autopsy) for every listed study. In addition, we have standardized the reporting of nanoparticle characteristics, now consistently including particle size (or size range), surface modification/functionalization, and exposure concentration and duration wherever available. These changes substantially improve the transparency, reproducibility, and comparability of the summarized studies.

  1. The review should address polymer degradation to oligomers, potential bio-corona composition changes over time, and the role of copolymerized or functionalized PS derivatives. Recent developments in biodegradable alternatives and advanced detection techniques should also be included to make the review forward-looking.

Thank you for your valuable comments. Please find our responses below.

A brief summary of the physical, chemical, and biological processes involved in plastic degradation to oligomers and monomers was provided in the Introduction. In this section, we also discuss the advancements in biodegradable alternatives to conventional plastics. We do not discuss the issue of advanced detection techniques of nanoplastics, as this topic is widely described in the literature.

The formation of a bio-corona on nanoplastics is a highly dynamic and time-dependent process. The composition changes of the bio-corona after entering the human body through ingestion and cellular uptake are provided in Section 2.1.

Several sentences are grammatically awkward or overly long; professional English editing is needed for fluency and conciseness.

Thank you for this comment. We have carefully revised the entire manuscript to improve clarity and fluency, and the full text has been checked and corrected.

Reviewer 2 Report

Comments and Suggestions for Authors

The review by the respected Agata Kustra and co-authors addresses the important topic of the mechanisms of nanoplastic (particularly polystyrene) effects on human gastrointestinal cells. Currently, there are many studies dedicated to microplastics due to increasing pollution levels; however, most of these focus on toxicity in animal models, especially aquatic ones. Significantly less is known about the cellular mechanisms of action of not micro-, but nanoplastics on humans. The relevance of the topic is driven, firstly, by the growing volume of nanoplastics in the environment and the fact that the main route of nanoplastic entry into the body is oral. On the other hand, cellular mechanisms of nanoplastic effects on humans and animals are only beginning to be studied. Thus, the review raises a timely and urgent issue.
Recently, reviews with similar topics have been published, but they either focused on different cell types (DOI: 10.3390/toxics12120908) or on different mechanisms (DOI: 10.1038/s41467-025-59884-y). Therefore, the review presented by the authors is novel and original.
The review is clearly structured; the text is logically presented and easy to understand. The review contains seven tables summarizing the results of key sections. It includes 116 references, 70% (81) of which are from the last five years. Thus, the review contains the most up-to-date information. The conclusions drawn by the authors are sound and supported by citations.
I highly value the authors' work and believe it must be published, while I have several suggestions I propose for discussion by the authors.
1. In the introduction, I would suggest adding a few sentences about what types of food-grade plastics exist and possibly indicating the volume of plastics produced, to clarify the choice of polystyrene. Specifically, six food plastics are currently actively used, so it would be helpful to justify why polystyrene was chosen for the review.
2. I am a bit puzzled by the mismatch between the title and some of the sections. The title refers to gastrointestinal models, while the authors discuss inhalation and transcutaneous routes in depth. If the authors want to cover all possible routes of nanoplastic entry into the body, there is also the parenteral route (from medical plastic systems and artificial vessels). Perhaps it would be better to remove these sections or combine points 2.2 and 2.3 as other routes of entry, concluding that the oral route is indeed the main one and deserves the greatest focus.
3. From the article, I would like a clearer conclusion on how harmful polystyrene nanoplastics can be to human health, especially in comparison to other nanoparticles, such as natural nanoparticles. Is nanoplastic truly a real health threat? Do polystyrene nanoparticles indeed cause deep damage to cellular viability?
4. Minor comment – please include explanations of all abbreviations in the footnotes of the tables.
5. Minor comment – please check the completeness of reference formatting, for example, reference 81 is ahead-of-print (2022) but should already be published. References 73, 66, 55 lack publication details, and reference 65 is questionable.

Author Response

Response to Reviewer 2: Comments – Round 1:

We are very thankful for the in-depth review of our manuscript. The remarks made by the Reviewer are very valuable, and we believe that implementing them has greatly improved the paper. Thank you for your time and constructive feedback.

All changes in the manuscript were marked in yellow.

The review by the respected Agata Kustra and co-authors addresses the important topic of the mechanisms of nanoplastic (particularly polystyrene) effects on human gastrointestinal cells. Currently, there are many studies dedicated to microplastics due to increasing pollution levels; however, most of these focus on toxicity in animal models, especially aquatic ones. Significantly less is known about the cellular mechanisms of action of not micro-, but nanoplastics on humans. The relevance of the topic is driven, firstly, by the growing volume of nanoplastics in the environment and the fact that the main route of nanoplastic entry into the body is oral. On the other hand, cellular mechanisms of nanoplastic effects on humans and animals are only beginning to be studied. Thus, the review raises a timely and urgent issue.
Recently, reviews with similar topics have been published, but they either focused on different cell types (DOI: 10.3390/toxics12120908) or on different mechanisms (DOI: 10.1038/s41467-025-59884-y). Therefore, the review presented by the authors is novel and original.
The review is clearly structured; the text is logically presented and easy to understand. The review contains seven tables summarizing the results of key sections. It includes 116 references, 70% (81) of which are from the last five years. Thus, the review contains the most up-to-date information. The conclusions drawn by the authors are sound and supported by citations.
I highly value the authors' work and believe it must be published, while I have several suggestions I propose for discussion by the authors.

  1. In the introduction, I would suggest adding a few sentences about what types of food-grade plastics exist and possibly indicating the volume of plastics produced, to clarify the choice of polystyrene. Specifically, six food plastics are currently actively used, so it would be helpful to justify why polystyrene was chosen for the review.

Thank you very much for this comment. Further explanation has been added in the “Introduction” section. It has been marked in yellow.

  1. I am a bit puzzled by the mismatch between the title and some of the sections. The title refers to gastrointestinal models, while the authors discuss inhalation and transcutaneous routes in depth. If the authors want to cover all possible routes of nanoplastic entry into the body, there is also the parenteral route (from medical plastic systems and artificial vessels). Perhaps it would be better to remove these sections or combine points 2.2 and 2.3 as other routes of entry, concluding that the oral route is indeed the main one and deserves the greatest focus.

We thank the Reviewer for this important and constructive remark. In response, we revised the structure and focus of Section 2 to better align it with the gastrointestinal scope of the manuscript. Sections 2.2 (respiratory exposure) and 2.3 (dermal exposure) were merged into a single concise subsection entitled “Other exposure routes”, clearly distinguished from the main gastrointestinal pathway. Their content was significantly shortened and is now presented only as complementary background. The oral route is now consistently emphasized as the dominant and most relevant pathway throughout the manuscript. The parenteral route was not included, as the focus of this review remains environmental exposure rather than medical applications. This restructuring eliminates the previous mismatch between the title and section content.

  1. From the article, I would like a clearer conclusion on how harmful polystyrene nanoplastics can be to human health, especially in comparison to other nanoparticles, such as natural nanoparticles. Is nanoplastic truly a real health threat? Do polystyrene nanoparticles indeed cause deep damage to cellular viability?

Thank you for your valuable comments. Please find our responses below.

Current evidence indicates that polystyrene nanoplastics (PS-NPs) constitute a genuine potential health risk. They can cross biological barriers, accumulate in tissues, and interact with cellular membranes and organelles. Their chemical stability, hydrophobicity, and ability to adsorb environmental pollutants further increase their toxic potential compared with many other nanoparticle types.

Natural nanoparticles (such as mineral or biopolymer-derived particles) generally exhibit lower toxicity due to their biodegradability, reduced surface reactivity, and limited bioaccumulation. In contrast, PS-NPs are synthetic, persistent, and display higher surface activity, which promotes cellular uptake and interaction with biomolecules. Their ability to act as carriers for co-contaminants gives them a distinct and often more harmful toxicological profile.

Numerous studies have demonstrated that PS-NPs induce significant cytotoxicity, including oxidative stress, mitochondrial dysfunction, DNA damage, membrane destabilization, and activation of apoptotic or necroptotic pathways. These mechanisms collectively contribute to reduced cellular viability across various cell types.

To address the Reviewer’s concern, a concise summary of these points has been added to the “Conclusions” section. It has been marked in yellow.

  1. Minor comment – please include explanations of all abbreviations in the footnotes of the tables.

We thank the Reviewer for this stylistic suggestion. In the revised manuscript, all abbreviations used in the tables are consistently explained in a dedicated list of abbreviations at the end of the manuscript. Given that many abbreviations recur across multiple tables, we believe that this centralized format significantly improves clarity, avoids unnecessary repetition, and enhances the overall graphical consistency and visual readability of the tables. This layout is also commonly used in review articles with extensive comparative tabular content.
5. Minor comment – please check the completeness of reference formatting, for example, reference 81 is ahead-of-print (2022) but should already be published. References 73, 66, 55 lack publication details, and reference 65 is questionable.

All references indicated by the Reviewer as incomplete, ahead-of-print, or questionable (including references 81, 73, 66, and 55) have been carefully verified. During the revision process, these references were either updated with full publication details where available or removed and replaced with fully published, peer-reviewed articles. The reference list has now been thoroughly checked to ensure completeness, accuracy, and consistency with the journal’s formatting requirements.

Round 2

Reviewer 1 Report

Comments and Suggestions for Authors

This manuscript is acceptable.

Reviewer 2 Report

Comments and Suggestions for Authors The authors have effectively addressed my comments and revised the manuscript thoroughly. I am fully satisfied with the changes made. I recommend acceptance of the article for publication.